# Clinical Phenotype of *PDE6B*-Associated Retinitis Pigmentosa

**DOI:** 10.3390/ijms22052374

**Published:** 2021-02-27

**Authors:** Laura Kuehlewein, Ditta Zobor, Katarina Stingl, Melanie Kempf, Fadi Nasser, Antje Bernd, Saskia Biskup, Frans P.M. Cremers, Muhammad Imran Khan, Pascale Mazzola, Karin Schäferhoff, Tilman Heinrich, Tobias B. Haack, Bernd Wissinger, Eberhart Zrenner, Nicole Weisschuh, Susanne Kohl

**Affiliations:** 1Institute for Ophthalmic Research, Centre for Ophthalmology, Eberhard Karls University Tübingen, 72076 Tübingen, Germany; Ditta.Zobor@uni-tuebingen.de (D.Z.); Fadi.Nasser@med.uni-tuebingen.de (F.N.); ez@uni-tuebingen.de (E.Z.); 2University Eye Hospital, Centre for Ophthalmology, Eberhard Karls University Tübingen, 72076 Tübingen, Germany; Katarina.Stingl@med.uni-tuebingen.de (K.S.); Melanie.Kempf@med.uni-tuebingen.de (M.K.); tuebingen@medizentrum-eckert.de (A.B.); 3Center for Rare Eye Diseases, Eberhard Karls University Tübingen, 72076 Tübingen, Germany; 4CeGaT GmbH, 72076 Tuebingen, Germany; Saskia.Biskup@humangenetik-tuebingen.de; 5Department of Human Genetics, Radboud University Medical Center, 6500 HB Nijmegen, The Netherlands; Frans.Cremers@radboudumc.nl (F.P.M.C.); mimranmani@gmail.com (M.I.K.); 6Donders Institute for Brain, Cognition and Behavior, Radboud University Medical Center, 6525 AJ Nijmegen, The Netherlands; 7Institute for Medical Genetics and Applied Genomics, University Hospital, Eberhard Karls University Tübingen, 72076 Tübingen, Germany; Pascale.Mazzola@med.uni-tuebingen.de (P.M.); Karin.Schaeferhoff@med.uni-tuebingen.de (K.S.); Tilman.Heinrich@med.uni-tuebingen.de (T.H.); Tobias.Haack@med.uni-tuebingen.de (T.B.H.); 8Centre for Rare Diseases, Eberhard Karls University Tübingen, 72076 Tübingen, Germany; 9Molecular Genetics Laboratory, Institute for Ophthalmic Research, Centre for Ophthalmology, Eberhard Karls University Tübingen, 72076 Tübingen, Germany; Bernd.Wissinger@med.uni-tuebingen.de (B.W.); Nicole.Weisschuh@uni-tuebingen.de (N.W.); Susanne.Kohl@med.uni-tuebingen.de (S.K.); 10Werner Reichardt Centre for Integrative Neuroscience (CIN), Eberhard Karls University Tübingen, 72076 Tübingen, Germany

**Keywords:** retinitis pigmentosa, phosphodiesterase 6

## Abstract

In this retrospective, longitudinal, observational cohort study, we investigated the phenotypic and genotypic features of retinitis pigmentosa associated with variants in the *PDE6B* gene. Patients underwent clinical examination and genetic testing at a single tertiary referral center, including best-corrected visual acuity (BCVA), kinetic visual field (VF), full-field electroretinography, full-field stimulus threshold, spectral domain optical coherence tomography, and fundus autofluorescence imaging. The genetic testing comprised candidate gene sequencing, inherited retinal disease gene panel sequencing, whole-genome sequencing, and testing for familial variants by Sanger sequencing. Twenty-four patients with mutations in *PDE6B* from 21 families were included in the study (mean age at the first visit: 32.1 ± 13.5 years). The majority of variants were putative splicing defects (8/23) and missense (7/23) mutations. Seventy-nine percent (38/48) of eyes had no visual acuity impairment at the first visit. Visual acuity impairment was mild in 4% (2/48), moderate in 13% (6/48), and severe in 4% (2/48). BCVA was symmetrical in the right and left eyes. The kinetic VF measurements were highly symmetrical in the right and left eyes, as was the horizontal ellipsoid zone (EZ) width. Regarding the genetic findings, 43% of the *PDE6B* variants found in our patients were novel. Thus, this study contributed substantially to the *PDE6B* mutation spectrum. The visual acuity impairment was mild in 83% of eyes, providing a window of opportunity for investigational new drugs. The EZ width was reduced in all patients and was highly symmetric between the eyes, making it a promising outcome measure. We expect these findings to have implications on the design of future *PDE6B*-related retinitis pigmentosa (RP) clinical trials.

## 1. Introduction

Retinitis pigmentosa (RP) is a hereditary degenerative retinal disease that causes severe visual impairment and vision loss due to the progressive degeneration of, primarily, the rod and, secondarily, the cone photoreceptors (i.e., rod–cone dystrophy). The disease manifests with early-onset nyctalopia followed by daytime visual field (VF) defects progressing from the mid-periphery to the periphery and the center. The best-corrected visual acuity (BCVA) typically remains relatively preserved until macular involvement by the macular edema, and/or photoreceptor atrophy causes central vision loss.

The rod photoreceptor cyclic guanosine monophosphate (cGMP) phosphodiesterase (PDE6) plays a crucial role in vertebrate phototransduction. The rod photoreceptor cGMP PDE comprises four subunits: one catalytic alpha-subunit PDE6A, one catalytic beta-subunit PDE6B, and two inhibitory gamma-subunits PDE6G. The enzyme functions to hydrolyze the intracellular cytoplasmic cGMP level, which causes the closure of cyclic nucleotide-gated channels upon illumination [1]. The gene encoding the beta subunit of PDE6 *(PDE6B)* was one of the first genes identified as causing retinal degeneration in mice, dogs, and humans (i.e., autosomal-recessive RP and autosomal-dominant stationary night blindness) [2,3,4,5,6,7,8,9,10,11]. The gene is located on chromosome 4p16.3 (OMIM *180072) and comprises 22 coding exons and encodes 854 amino acid residues [12].

Treatment options for *PDE6B*-associated RP are currently not available, thought they are under investigation. Proof-of-concept studies with subretinal gene therapy showed positive effects in mice and dogs when treated at a very early time during postnatal development of the retina [13,14,15,16]. A clinical trial on the safety and efficacy of gene therapy in human patients with RP caused by biallelic mutations in the *PDE6B* gene is ongoing (ClinicalTrials.gov Identifier: NCT03328130). More recently, the transplantation of chemically induced photoreceptor-like cells (CiPCs) into the subretinal space of *rd1* mice, which are a homozygous mutant for *Pde6b*, showed a partial restoration of the pupil reflex and visual function [17]. For the success of therapeutic interventions, it is necessary to carefully study the natural course and specific hallmarks of the disease.

The aim of the present study was to analyze the clinical features and genetic findings of patients with RP caused by biallelic mutations in the *PDE6B* gene examined at a single clinical site, the Centre for Ophthalmology of the University of Tübingen, Germany.

## 2. Results

Of the 24 patients (48 eyes) included in this analysis, 46% (11/24) were female. The mean age at the first visit was 32.1 ± 13.5 years (range, 7–52 years). Follow-up data was available for 67% (16/24) of the patients. The mean follow-up time was 6.9 ± 2.9 years (range, 1–11 years). The disease onset was reported most frequently during the patients’ first (*n* = 16) or second (*n* = 5) decades of life. One patient reported her first symptoms during her third decade of life (SRP759-19456).

### 2.1. Genetic Findings

All 24 cases harbored one homozygous or two heterozygous rare and potentially disease-causing variants in the *PDE6B* gene, compatible with an autosomal-recessive mode of inheritance (Table 1). Twenty-three distinct *PDE6B* variants were observed (Figure 1). Thirteen of the variants have already been reported in the literature in association with an inherited retinal disease (IRD), mostly RP, and ten were novel. All variants were classified according to the American College of Medical Genetics and Genomics (ACMG) guidelines (Table 2). Apparent homozygous variants were observed in 15 and two heterozygous variants in nine patients. In the homozygous cases, there was no evidence of copy number variations in the genome or panel sequencing analyses. True biallelic genotypes were established in six patients (four families) by segregation analysis in both parents (Table 1). Out of the nine subjects (seven families) with two heterozygous variants, segregation analysis was performed in four patients (three families), unequivocally confirming the biallelic genotypes. Consequently, in five cases, biallelic mutations were not confirmed by segregation analysis.

The mutation spectrum comprises eight putative splice site variants, of which six affect the canonical splice acceptor or donor sequences, seven missense and four nonsense variants, and two small (one in-frame and one out-of-frame) and one larger duplication, as well as one frame-shifting indel variant (Figure 1 and Table 2). The only true recurrent alleles were the nonsense variant c.892C>T;p.(Q298*), accounting for eight disease-associated alleles in six patients, twice homozygously, and the missense variant c.2326G>A;p.(D776N), accounting for five alleles in three patients, two of them being apparently homozygous for this variant (Table 2).

### 2.2. Ophthalmological Findings

The detailed ophthalmological findings are provided in Table 1. The mean BCVA at the first visit was 0.3 ± 0.3 logMAR (range, −0.1–1.2, *n* = 48). Seventy-nine percent (38/48) of eyes had no visual acuity impairment. The visual acuity impairment at the first visit was mild in 4% (2/48), moderate in 13% (6/48), and severe in 4% (2/48), as defined by the World Health Organization (WHO)—that is, mild-presenting visual acuity worse than 6/12, moderate-presenting visual acuity worse than 6/18, severe-presenting visual acuity worse than 6/60, and blindness-presenting visual acuity worse than 3/60. The BCVA was symmetrical in the right and left eyes (R^2^ = 0.604). The BCVA with respect to age/disease duration is shown in Figure 2A.

The kinetic VF testing obtained with target III4e of Goldmann allowed for an evaluation in 96% (46/48) of the eyes at the first visit. Thirty-three percent (16/48) had a VF area greater than the equivalent of a concentric VF with a diameter of 60° (30° each from the center). Accordingly, 15% (7/48) of the eyes had a VF area with a diameter of 40°–60°, 23% (11/48) of eyes had a VF area with a diameter of 20°–40°, and 25% (12/48) of eyes had a VF area within 10° each from the center. The kinetic VF testing obtained with target I4e of Goldmann allowed for an evaluation in 85% (41/48) of the eyes. The kinetic VF measurements were highly symmetrical in the right and left eyes (R^2^ = 0.906 for target III4e and R^2^ = 0.979 for target I4e). The progression of VF defects with respect to the age/disease duration is shown in Figure 2B,C.

Optical coherence tomography (OCT) images were available in 92% (44/48) of the eyes. As typical for RP, OCT imaging revealed thinning of the outer retinal layers from the periphery to the center of the retina (macula/fovea), with disruption or loss of the ellipsoid zone (EZ) in the advanced disease. The qualitative findings and quantitative measures of OCT parameters, i.e., horizontal EZ width and foveal thickness (FT), are provided in Table 3. The horizontal EZ width was highly symmetrical in the right and left eyes (Table 3). Yet, in 21% (10/48) of the eyes, the horizontal EZ width was not distinguishable due to advanced disease.

There was no statistically significant correlation between the EZ width and VF area using target III4e (*p* = 0.118 for right eyes, *N* = 15 and *p* = 0.296 for left eyes, *N* = 14); however, there was a statistically significant correlation between the EZ width and VF area using target I4e (Spearman’s rho (r_s_) = 0.853, *p*<0.01 for right eyes, *N* = 12 and r_s_ = 0.636, *p* = 0.035 for left eyes, *N* = 11).

Fundus autofluorescence (FAF) images were available in 92% (44/48) of the eyes. Common findings were a parafoveal hyperautofluorescent ring in 73% (35/48) of the eyes, decreased autofluorescence in the mid-periphery in 73% (35/48) of the eyes, and patches of well-demarcated definitely decreased autofluorescence (i.e., patches of atrophy) along the arcades in 50% (24/48) of the eyes.

Full-field electroretinography (ff-ERG) findings were available in 73% (35/48) of the eyes. In 52% (25/48), dark- and light-adapted ff-ERG responses were absent. In 19% (9/48), ff-ERG showed residual/subnormal light-adapted responses, whereas dark-adapted ff-ERG showed no responses. In 4% (2/48), ff-ERG showed responses in dark- and light-adapted conditions. The age range of patients with responses in ff-ERGs was 9–53 years. Full-field stimulus threshold (FST) findings using white stimuli were available in 46% (11/24) of the patients. The median FST was −16.1 dB (interquartile range, 5.4 dB).

The most frequent variant observed in our cohort was c.892C>T;p.(Q298*), present in a homozygous state in two unrelated patients (8%, 2/24). The phenotype of these unrelated patients with respect to the BCVA and VF area with target III4e was similar to that of other patients in the same age group with different genotypes (see Figure 2B for the VF area and Figure 3). The second-most frequent variant, c.2326G>A;p.(D776N), was also present in a homozygous state in two patients (8%, 2/24). The patients were also unrelated. The BCVA values were among the best when compared to patients in the same age group, and the VF area with target III4e was even better in comparison to patients in the same age group with different genotypes (see Figure 2B and Figure 3), indicating a milder phenotype associated with this very mutation and genotype.

## 3. Discussion

In this study, we reported 24 patients from 21 families with RP caused by likely pathogenic and biallelic variants in the *PDE6B* gene. The patients were part of the Tübingen cohort of >7700 non-syndromic IRD index cases, of which 1653 had a clinical diagnosis of sporadic or suspected autosomal-recessive RP and were tested for mutations in *PDE6B*. Within this cohort, 30 individuals (of which 24 were included in this study) likely carried biallelic *PDE6B* variants, indicating that 0.4% of all IRD cases in our cohort and 1.8% of sporadic or autosomal-recessive RP cases in our cohort were caused by mutations in *PDE6B*. Thus, we concluded the frequency of *PDE6B*-associated IRD in Germany to be lower than that reported for France (2.4% in a cohort of patients with rod-cone dystrophy) and North America (4% in a cohort of patients with autosomal-recessive RP) [26,27]. The youngest patient in our cohort was seven and the oldest patient 52 years old at the first visit. Although 80% of the eyes (*n* = 48) had no visual acuity impairment, 60% of the eyes had severe VF defects, with a VF area smaller than the equivalent of a concentric VF with a diameter of 60° (30° each from the center).

### 3.1. Genetic Findings

Mutations were identified in all but one case by comprehensive genetic testing analyzing all the genes associated with RP or with inherited retinal dystrophies in general. The fact that no other likely disease-associated genotypes were identified by these analyses supports the association of the presented *PDE6B* genotypes with the RP phenotype in these cases, although a segregation analysis was lacking for the majority of our patients. We consider the homozygous cases to be truly homozygous, as there was no evidence of copy number variations in the panel or genome sequencing analyses.

The majority of variants were putative splicing defects (8/23) and missense (7/23) mutations. All the splice site variants were predicted to result in erroneous splicing, and all the missense variants affected the amino acid residues that are not only conserved among the PDE6B orthologs of various vertebrate species but, also, in the paralogous rod photoreceptor PDE6A and cone photoreceptor PDE6C (Appendix A) and are located in the functional domains of the PDE6B protein (Figure 1). Yet, according to the ACMG guidelines, all the missense variants are classified as variants of uncertain significance, since functional studies on the effects of these variants are lacking. Four nonsense variants leading to premature termination codons were observed, leading to severely truncated and likely nonfunctional proteins, but all were also predicted to target the mutant transcripts to nonsense-mediated decay.

Two complex alleles were observed, homozygous in each case. The allele c.[299G>A;1401+4_1401+48del] gives rise to a missense variant and a putative splice defect (p.[(R100H);(?)] and was seen in two siblings (family ARRP209). Both variants have been reported before in the literature in association with autosomal-recessive RP but not as complex alleles [22,28]. The other complex allele c.[409G>A;928-9_940dup] was seen in a single patient and again represents a missense variant and a 22-bp duplication, covering the last nine nucleotides of intron 5 and the first 13 nucleotides of exon 6 and, potentially, also affecting the splicing (p.[(G137R);(?)]). Again, both variants have been described in the literature in association with autosomal-recessive RP and stationary night blindness, respectively [19,20]. Yet, it has to be noted that the frequency of the two putative splice variants c.928-9_940dup and c.1401+4_1401+48del in the general population (minor allele frequencies 0.5% and 3.56%, respectively, and, especially, high in non-Finnish Europeans (0.74% and 5.72%, respectively)) may indicate that these need to be considered as likely benign variants. In contrast, the two missense variants within these complex alleles, c.299G>A;p.(R100H) and c.409G>A;p.(G137R), are very likely deleterious, both affecting highly conserved amino residues in the GAF1 domain of PDE6B, PDE6A, and PDE6C. In addition, variant c.299G>A;p.(R100H) affects an arginine residue corresponding to the known disease-associated recurrent variants in *PDE6A* (p.R102) and *PDE6C* (p.R104), suggesting that this arginine codon represents a mutation hotspot [29]. Therefore, we propose that these are likely the true pathogenic mutations associated with the *PDE6B*-RP phenotype in our patients.

### 3.2. Clinical Findings

In our cohort, the mean BCVA was better than that reported by Khateb et al. (0.3 logMAR vs 0.4 logMAR), which may be explained by the age distribution in both cohorts (mean age 32.1 years vs. 41.6 years (*n* = 35)) but, also, other factors such as additional macular pathology [27]. The BCVA findings between the right and left eyes were less symmetrical when compared to kinetic VF findings obtained with targets III4e and I4e, which may be explained by the fact that the BCVA was recorded in logMAR units, whereas the VF area was recorded in square degrees, which allows for more finely graded measurements. Additionally, 7/24 patients in our cohort exhibited asymmetry in the qualitative OCT findings with respect to the foveal region, e.g., cystoid macular edema that was more pronounced in one eye or a lamellar macular hole with foveal EZ disruption in one eye, respectively. The symmetry in the VF was less pronounced in eyes with large(r) VF areas, which may explain that the VF findings were more symmetrical when obtained with target I4e (as the VF area obtained with target I4e will typically be smaller than that obtained with target III4e). Yet, in 15%, target I4e was not recognized, which provides a case for using both targets in VF testing in clinical studies. Of note, our VF data reflected the natural course of RP, characterized by a loss of large(r) VF areas in young adulthood and the subsequent slow progression of VF defects the smaller the remaining VF area gets (Figure 2).

The horizontal EZ width on OCT imaging has been shown to closely correlate with the VF area [30]. The strong and statistically significant correlation between the EZ width and VF area obtained with target I4e in our study—while no statistically significant correlation was observed between the EZ width and VF area obtained with target III4e—may be explained by the absence of peripheral islands in the VF measurements obtained with target I4e. Additionally, symmetrical progressive loss of the EZ line width was shown for patients harboring disease-causing variants in either *PDE6A* or *PDE6B* [31]. Although the standard OCT captures only the posterior pole of the eye, the EZ integrity did not exceed the scan area in any of the eyes, even in the youngest patients with the least advanced disease. The mean horizontal EZ width was less than that reported by Khateb et al. (2042 µm vs. 2198 µm in the right eyes and 1924 µm vs. 2178 µm in the left eyes) [27]. The Symmetry in the horizontal EZ width was more pronounced when compared to the VF area obtained with target III4e. In our study, two-thirds of the patients exhibited an epiretinal membrane, which was more than that reported by Khateb et al. (one-third), and around one-third of the patients exhibited cystoid macular edema, which is comparable to that reported by Khateb et al. [27].

Disease symmetry presents a strong argument for using the untreated eye as the control eye in an interventional study with small patient numbers. The untreated eye may serve as a control to assess the possible negative side effects, as well as positive effects, of (e.g., gene supplementation) therapy (i.e., the preservation of the VF area or the deceleration of the VF area loss). Given the remarkable symmetry of the VF area and horizontal EZ width in the right and left eyes that we observed in our cohort, we judged both measures as suitable endpoints in clinical trials with some limitations. Kinetic VF testing is a psychophysical exam. The condition of the patient impacts the results of the exam. Thus, VF findings can vary between visits and comparisons between patients, as well as within the same patient at different time points, wich may yield misleading results. Additionally, if a patient participates in a clinical trial testing an investigational new drug and the patient needs to undergo surgery, he/she will know which eye is the treated eye and which eye serves as the untreated control, which may well impact the results of the VF testing as well. The EZ width is an anatomical parameter that cannot be modified by the patient. Yet, in 21% of eyes, the EZ width was not distinguishable due to advanced disease, whereas the VF testing obtained with target III4e allowed for an evaluation in 96% of the eyes.

Besides the study conducted by Khateb et al. in which the authors collected the retrospective data of 35 *PDE6B*-associated RP patients of 26 families, several cases and case series have been reported in the literature [24,27,32,33,34,35,36]. Their phenotypes were described as typical RP, which is in line with our findings [27,32]. More specifically, Khateb et al. described *PDE6B*-associated RP as a classic autosomal-recessive RP with night blindness and progressive visual field constriction but with relatively preserved central vision (i.e., macular cones) at older ages, which is again in line with our findings [27].

None of the patients exhibited clinical features of congenital stationary night blindness, i.e., an electronegative ERG. Of note, none of the patients exhibited the heterozygous c.772C>A;p.(H258N) variant, which has been associated with autosomal-dominant stationary night blindness [10].

Patients homozygous for the second-most frequent variant in our cohort, the missense variant c.2326G>A;p.(D776N), exhibited a milder phenotype when compared to patients in the same age group with different genotypes, especially with respect to their VF. The variant leads to an amino acid exchange in the PDEase I catalytic domain and may result in a partial loss of the enzymatic function. To our knowledge, this variant has not been reported in the literature to be present in the homozygous state. Notably, we considered the size of our cohort too small and the genetic diversity too large for an actual genotype–phenotype correlation. Additionally, there may well be other factors (i.e., genetic and environmental) with an effect on the resulting phenotype. Yet, these findings are in line with the well-known *Pde6b* mouse models: *rd1* carries a large insertion followed by a nonsense mutation and most likely represents a *null* allele and is known for its fast degeneration. In contrast, the *rd10* model carries a missense variant (p.R560C) and is known for its milder retinal degeneration phenotype [37].

*PDE6B*-associated RP is a primary rod disease. If treated with gene supplementation, the clinical endpoints should include the evaluation of the rod function and rod structure. There has been no evidence of rod function and rod structure in patients published to date [24,27,32,33,34,35,36]. FST testing (where available) showed a threshold around −16 dB in most patients. This value is driven by cones and is “normal” for cones. Only when the cones start to degenerate does the threshold increase, which was the case for one patient in our study (ARRP209-23862). The responses in dark-adapted ff-ERG were detected in one patient in our cohort, obtained with 3 and 10-cds/m^2^ light flashes. Yet, we judged these responses to be driven by cones [38]. The VF area obtained with target III4e in this patient was considerably large in both eyes; yet, there were two patients with even larger III4e VF areas with absent dark-adapted ff-ERG responses. The patient did, however, have the largest VF area obtained with object I4e in our cohort (SRP612-17465; Figure 2C).

In our study, we described the detailed genetic and clinical findings in the second-largest cohort of patients with *PDE6B*-associated RP published to date. However, our study had limitations—in particular, due to its retrospective design. Given that PDE6 is rod-specific, it would be of interest to collect more detailed data on rod functions in *PDE6B*-associated RP, e.g., dark-adapted chromatic perimetry or pupil campimetry. Yet, the safety and efficacy measures in clinical trials (e.g., gene supplementation) will currently likely remain cone-driven measures such as the preservation of the VA/VF area or the deceleration of the VA/VF area loss or anatomical endpoints such as the EZ width.

In summary, we observed and described the genetic and ophthalmologic characteristics in 24 patients with RP caused by the likely pathogenic and biallelic variants in the *PDE6B* gene. Regarding the genetic findings, 43% (10/23) of the *PDE6B* variants found in our patients were novel. Thus, this study contributed substantially to the *PDE6B* mutation spectrum. Regarding the clinical findings, the disease was highly symmetrical between the right and left eyes, and the visual acuity impairment was mild in 83% of the eyes, providing a window of opportunity for investigating new drugs. As rod ERG was extinguished in most patients and FST was cone-driven, alternative methods of assessing the rod function are needed for future studies of this rod-driven disease. We expect these findings to have implications on the design of future *PDE6B*-related RP clinical trials.

## 4. Materials and Methods

The study was conducted in accordance with the Declaration of Helsinki, with approval from the ethics committee of the University of Tübingen (project no. 384/2020BO, 26 May 2020 and 116/2015BO2, 15 June 2018).

### 4.1. Ophthalmological Testing

Twenty-four patients from 21 families were included in the study. All patients were examined at the Centre for Ophthalmology of the University of Tübingen, Germany, a tertiary referral center. A comprehensive ophthalmological examination was performed, including BCVA, a semi-automated 90° kinetic VF exam with targets lll4e and I4e (Octopus 900, Haag-Streit, Wedel, Germany), ff-ERG testing according to the ISCEV standards (Espion, Diagnosys, Lowell, MA, USA), FST testing, spectral domain OCT and FAF imaging (Spectralis^®^ HRA+OCT, Heidelberg Engineering GmbH, Heidelberg, Germany), and slit-lamp and dilated fundus examination and photography. BCVA was converted to logMAR visual acuity [39]. VF parameters were assessed as the total VF area in square degrees for both targets III4e and I4e using standard preinstalled software of the device. Correlation between the EZ width and VF area was studied using Spearman’s rho (r_s_). Quantitative OCT parameters were assessed with the standard pre-installed software of the device (HEYEX, Heidelberg Engineering GmbH, Heidelberg, Germany). Horizontal width of the EZ and FT were measured as previously described [40]. For each eye, horizontal line scans through the fovea were analyzed. Firstly, the boundary lines for the outer limiting membrane, EZ, and proximal border of the retinal pigment epithelium were determined. Then, the location where the EZ merges with the proximal border of the retinal pigment epithelium was marked on either side (temporal and nasal to the fovea). Sixty-seven percent (16/24) of the patients were followed up on.

### 4.2. Genetic Testing and Variant Classification

Genetic testing comprised candidate gene sequencing (*n* = 1, ARRP171-15079), IRD gene panel sequencing (*n* = 19), whole-genome sequencing (*n* = 1, ARRP411-30491), or testing for familial variants by Sanger sequencing in a research and/or diagnostic genetic set-up [41,42,43]. A segregation analysis was performed on the parental samples available from six index cases to confirm a biallelic status of the identified *PDE6B* variants (Table 1). Variants were classified according to the standards and guidelines provided by the American College of Medical Genetics and Genomics (ACMG) and the Association for Molecular Pathology (AMP) [18]. The potential pathogenicity of the missense changes was further assessed by applying various prediction tools embedded in Alamut Visual software (Interactive Biosoftware, Rouen, France), literature research, and conservation between the catalytic subunits of the specific rod photoreceptors PDE6B and PDE6A, as well as the cone photoreceptor PDE6C. All variants were annotated according to the NCBI reference sequence for *PDE6B* (ENST00000496514, NM_000283.3, and NP_000274.2) comprising 22 coding exons. The novel variants were submitted to ClinVar (http://www.ncbi.nlm.nih.gov/clinvar/) under submission number SUB9063448.

## Figures and Tables

**Figure 1 ijms-22-02374-f001:**
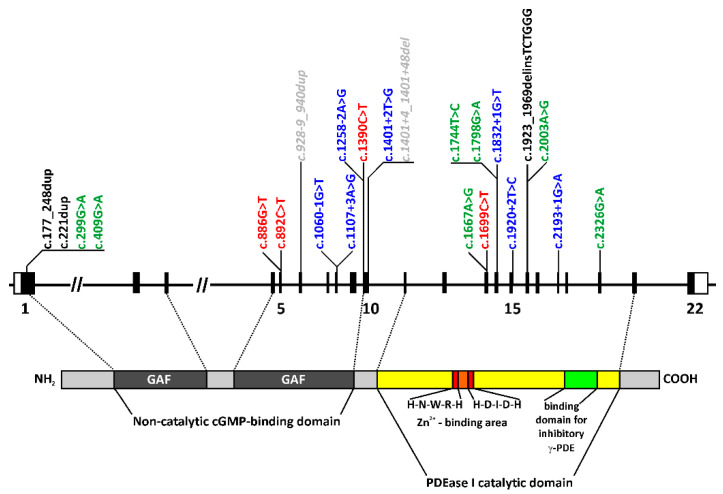
Genomic and protein structure of PDE6B and the locations of variants found in the patients of this study. The exon/intron organization of the *PDE6B* gene is shown to scale at the top, while the PDE6B polypeptide and its functional domains are shown below. The two functionally important GAF domains of the noncatalytic cyclic guanosine monophosphate (cGMP)-binding domain are located in the amino terminal half of the protein, and the highly conserved Zn^2+^-binding motifs and the binding domain for the inhibitory γ-subunit PDE6H are located within the PDEase I catalytic domain in the carboxyterminal half of the protein. Color code of the variants: red = nonsense variants, green = missense variants, blue = splice site variants, black = insertion/deletion and duplication variants, and grey = likely benign variants.

**Figure 2 ijms-22-02374-f002:**
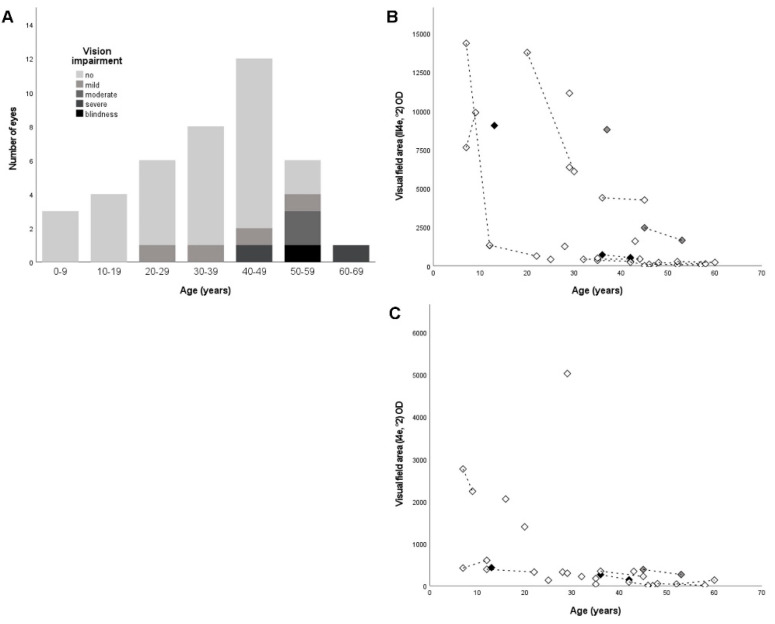
Visual acuity and kinetic visual field in *PDE6B*-associated retinitis pigmentosa. (**A**) Best-corrected visual acuity (BCVA) in right eyes (first visit and follow-up) with respect to age/disease duration. Notably, the BCVA was good until the end of the fourth decade in our cohort. (**B**) Kinetic visual field (VF) area obtained with target III4e of Goldmann in right eyes with respect to age/disease duration. Follow-up is indicated by the dashed lines. Note the large(r) loss of visual field (VF) area before the age of 30, which reflects the natural course of the disease. (**C**) VF area obtained with target I4e in right eyes with respect to age/disease duration. Follow-up is indicated by the dashed lines. Note the outlier, which is the patient in which dark-adapted 3.0 cds/m^2^ full-field electroretinography showed response (SRP-612-17465). Black symbols indicate patients homozygous for the most frequent variant in our cohort, c.892C>T;p.(Q298*), grey symbols indicate patients homozygous for the second most frequent variant in our cohort, c.2326G>A;p.(D776N), and white symbols indicate the other patients with different genotypes.

**Figure 3 ijms-22-02374-f003:**
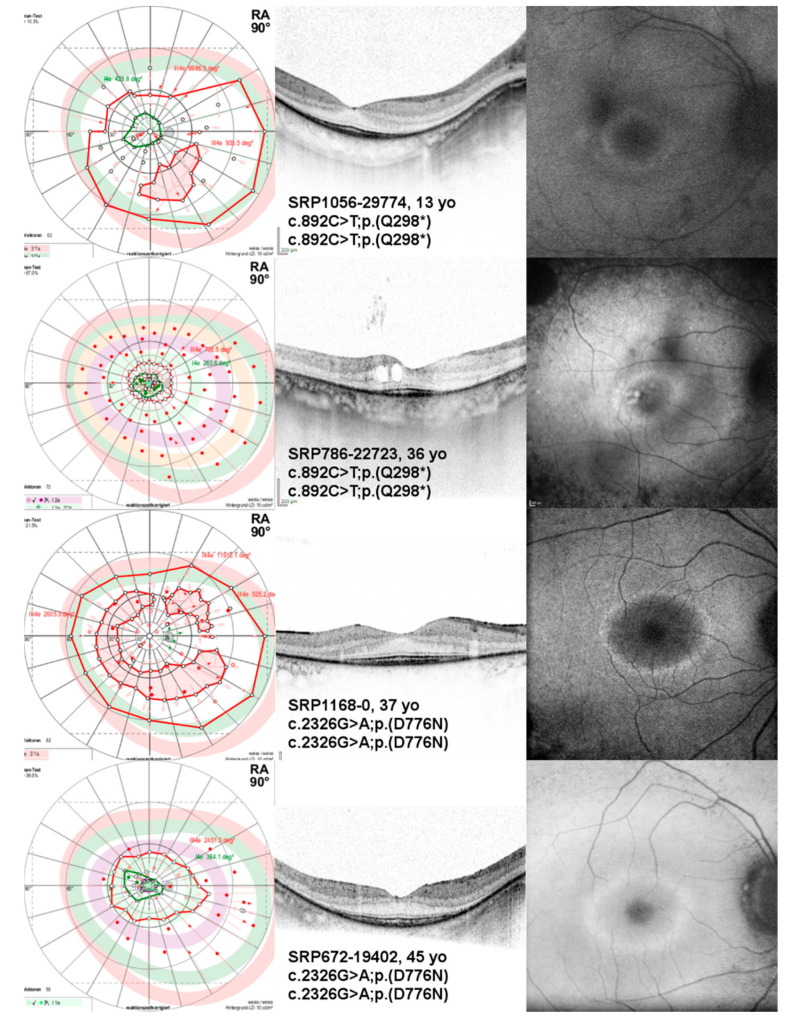
Kinetic visual field (VF), optical coherence tomography (OCT), and fundus autofluorescence (FAF) images of the right eyes of patients with *PDE6B*-associated retinitis pigmentosa harboring the variant c.892C>T;p.(Q298*) or variant c.2326G>A;p.(D776N) in a homozygous state. Note the difference in VFs despite the age of the patients. Additionally, note the thinning of the outer retinal layers outside the center of the OCT imaging, which is typical for retinitis pigmentosa (RP). Patient SRP786-22723 additionally exhibits cystoid macular edema. Note the parafoveal hyperautofluorescent ring on FAF imaging for all patients.

**Table 1 ijms-22-02374-t001:** Genetic and clinical findings in *PDE6B*-associated retinitis pigmentosa.

ID	Gender	Age	Disease Onset	Variant 1	Variant 2	BCVA (logMAR) OD/OS	VF (III4e, °^2^) OD/OS	VF (I4e, °^2^) OD/OS
LCA120-28043	M	7	1st decade	c.892C>T;p.(Q298*)	c.886G>T;p.(E296*)	0.1/0.1	7636/7651	2764/2345
ARRP260-25421§	M	7	1st decade	c.177_248dup;p.(L60_L83dup)	c.1401+2T>G;p.(?)	0.2/0.1	14359/9883	420/672
ARRP260-25423§	M	12	2nd decade	c.177_248dup;p.(L60_L83dup)	c.1401+2T>G;p.(?)	0.2/0.1	1325/1545	390/417
SRP1056-29774§	M	13	1st decade	c.892C>T;p.(Q298*)	c.892C>T;p.(Q298*)	0.0/-0.1	9051/8358	430/413
ARRP395-30213	F	16	2nd decade	c.1832+1G>T;p.(?)	c.1832+1G>T;p.(?)	0.3/0.2	NP/NP	2054/1502
SRP759-19456	F	20	3rd decade	c.1258-2A>G;p.(?)	c.2193+1G>A;p.(?)	0.2/0.4	1376/13149	1395/1221
ARRP411-30491	M	25	1st decade	c.1060-1G>T;p.(?)	c.1060-1G>T;p.(?)	0.0/0.0	412/493	131/162
ARRP75-5835§	M	28	1st decade	c.1699C>T;p.(Q567*)	c.1699C>T;p.(Q567*)	0.1/0.6	1249/1801	323/204
SRP1035-29494	M	29	1st decade	c.1107+3A>G;p.(?)	c.1920+2T>C;p.(?)	0.3/0.3	6354/10143	297/225
SRP612-17465	F	29	1st decade	c.1667A>G;p.(Y556C)	c.1667A>G;p.(Y556C)	0.3/0.3	11137/9484	5025/5129
SRP774-22406	F	32	Birth	c.1390C>T;p.(Q464*)	c.1390C>T;p.(Q464*)	0.1/0.2	416/440	218/277
ARRP26-21885§	M	35	1st decade	c.892C>T;p.(Q298*)	c.2003A>G;(p.D668G)	0.3/0.3	349/427	172/33
SRP823-26156	M	35	2nd decade	c.2193+1G>A;p.(?)	c.2193+1G>A;p.(?)	0.2/0.2	484/420	36/18
SRP786-22723	F	36	1st decade	c.892C>T;p.(Q298*)	c.892C>T;p.(Q298*)	0.1/0.0	709/588	264/53
SRP778-22500	F	36	1st decade	c.1923_1969delinsTCTGGG;p.(N643Gfs*29)	c.2326G>A;p.(D776N)	0.0/0.0	4388/7223	345/398
SRP1168-0	F	37	ND	c.2326G>A;p.(D776N)	c.2326G>A;p.(D776N)	-0.1/-0.1	8785/7893	NP/NP
ARRP209-23862	F	42	1st decade	c.[299G>A;1401+4_1401+48del];p.[(R100H);(?)]	c.[299G>A;1401+4_1401+48del];p.[(R100H);(?)]	0.3/0.6	228/239	88/66
ARRP26-18556§	F	43	2nd decade	c.892C>T;p.(Q298*)	c.2003A>G;(p.D668G)	0.2/0.7	1586/1579	341/398
SRP672-19402	M	45	1st decade	c.2326G>A;p.(D776N)	c.2326G>A;p.(D776N)	0.1/0.1	2451/2259	384/272
ARRP209-22048	M	45	1st decade	c.[299G>A;1401+4_1401+48del];p.[(R100H);(?)]	c.[299G>A;1401+4_1401+48del];p.[(R100H);(?)]	0.3/0.1	9/3	NP/NP
ARRP171-15079#	F	46	2nd decade	c.1798G>A;p.(D600N)	c.1798G>A;p.(D600N)	1.2/1.2	96/226	8/ND
SRP960-28509	F	48	1st decade	c.[409G>A;928-9_940dup];p.[(G137R);(?)]	c.[409G>A;928-9_940dup];p.[(G137R);(?)]	0.1/0.1	208/287	49/37
SRP341-24713	M	52	1st decade	c.1744T>C;p.(Y582H)	c.1744T>C;p.(Y582H)	0.6/0.4	54/34	NP/NP
SRP754-21728	M	52	1st decade	c.221dup;p.(V75Rfs*91)	c.892C>T;p.(Q298*)	1.0/0.6	269/185	40/23

Footnote: ARRP260-25421 and ARRP260-25423 are siblings, as are ARRP26-21885 and ARRP26-18556 and ARRP209-23862 and ARRP209-22048. § Segregation in both parents. # Segregation in offspring. Abbreviations: M = male, F = female, ND = no data, BCVA = best-corrected visual acuity, OD = right eye, OS = left eye, VF = (kinetic) visual field, and NP = not performed, °^2^ = square degrees. Description of sequence variation according to the recommentations of the Human Genome Variation Society (HGVS, http://varnomen.hgvs.org/).

**Table 2 ijms-22-02374-t002:** Mutation spectrum in *PDE6B* and the number of alleles in this study.

*PDE6B* cDNA (NM_000283.3)	PDE6B Protein (NP_000274.2)	Number of Alleles	Reference (dbSNP & Literature)	Location of Missense Variant in Protein Domain	Consensus Predictions of Various Software §	gnomAD MAF	ACMG Prediction [18]	ACMG Classification [18]
c.177_248dup	p.(L60_L83dup)	2	This study		in-frame duplication	n.d.	Pathogenic (IIIa)	PM2, PM4, PP1, PM3
c.221dup	p.(V75Rfs*91)	1	This study		frame-shift, PTC, NMD	n.d.	Likely pathogenic (I)	PM2, PVS1
c.299G>A	p.(R100H)	4	rs555600300 [12]	GAF1	damaging	0.0057%	VUS	PM2
c.409G>A	p.(G137R)	2	rs781658083 [19]	GAF1	damaging	0.00092%	VUS	PM2
c.886G>T	p.(E296*)	1	rs1064797304 This study		PTC, NMD	n.d.	Likely pathogenic (I)	PM2, PVS1
c.892C>T	p.(Q298*)	8	rs121918579 [5]		PTC, NMD	0.005%	Pathogenic (Ib)	PM2, PVS1, PM3
c.928-9_940dup	p.(?)	2	rs539992414 [20]		benign	0.50%	Likely pathogenic (I)	PM2, PVS1
c.1060-1G>T	p.(?)	2	rs863223339 [21]		missplicing	n.d.	Likely pathogenic (I)	PM2, PVS1
c.1107+3A>G	p.(?)	1	rs370898371 [12]		missplicing	0.0042%	VUS	PM2
c.1258-2A>G	p.(?)	1	This study		missplicing	n.d.	Likely pathogenic (I)	PM2, PVS1
c.1390C>T	p.(Q464*)	2	This study		PTC, NMD	n.d.	Likely pathogenic (I)	PM2, PVS1
c.1401+2T>G	p.(?)	2	This study		missplicing	n.d.	Pathogenic (Ia)	PM2, PVS1, PP1, PM3
c.1401+4_1401+48del	p.(?)	4	rs768567008 [22]		possibly missplicing	3.56%	VUS	PM2
c.1667A>G	p.(Y556C)	2	rs755577875 This study	PDEase I catalytic domain	damaging	0.001%	VUS	PM2
c.1699C>T	p.(Q567*)	2	rs772057239 [23]		PTC, NMD	0.0011%	Pathogenic (Ib)	PM2, PVS1, PM3
c.1744T>C	p.(Y582H)	2	This study	PDEase I catalytic domain	damaging	n.d.	VUS	PM2
c.1798G>A	p.(D600N)	2	rs764605140 [24]	PDEase I catalytic domain second Zn^2+^ binding motif	damaging	0.0056%	VUS	PM2
c.1832+1G>T	p.(?)	2	This study		missplicing	n.d.	Likely pathogenic (I)	PM2, PVS1
c.1920+2T>C	p.(?)	1	rs763996159 [12]		missplicing	0.0008%	Likely pathogenic (I)	PM2, PVS1
c.1923_1969delinsTCTGGG	p.(N643Gfs*29)	1	[25]		frame-shift, PTC, NMD	n.d.	Likely pathogenic (I)	PM2, PVS1
c.2003A>G	(p.D668G)	2	This study	PDEase I catalytic domain	damaging	n.d.	VUS	PM2, PM3
c.2193+1G>A	p.(?)	3	rs727504075 [26]		missplicing	0.0072%	Likely pathogenic (I)	PM2, PVS1
c.2326G>A	p.(D776N)	5	rs141563823 [12]	PDEase I catalytic domain	damaging	0.0057%	VUS	PM2

Abbreviations: gnomAD (Genome Aggregation Database: https://gnomad.broadinstitute.org/), PTC, premature termination codon, and NMD, nonsense-mediated decay. Footnote: § The potential pathogenicity of missense changes was assessed applying various prediction tools embedded in the Alamut Visual software (Interactive Biosoftware, Rouen, F), i.e., phyloP; Grantham distance; Align GVGD; SIFT; Mutationtaster; PolyPhen-2 for missense variants; and MaxENT, NNSPLICE, and SSF for splice site variants. Description of sequence variation according to the recommendations of the Human Genome Variation Society (HGVS, http://varnomen.hgvs.org/).

**Table 3 ijms-22-02374-t003:** Qualitative findings and quantitative measures of optical coherence tomography parameters in *PDE6B*-associated retinitis pigmentosa (RP).

	Frequency	N						
ERM	67%	32/48						
CME	35%	17/48						
Macular atrophy	19%	9/48						
Posterior staphyloma associated with myopia	10%	5/48						
Papillomacular retinal thickening	8%	4/48						
IHRM	6%	3/48						
Lamellar macular hole	2%	1/48						
		**N**	**Median**	**Mean**	**SD**	**Min**	**Max**	**Symmetry**
EZ-width OD (µm)		18/24	2042	2042	874	696	3810	R^2^ = 0.942
EZ-width OS (µm)		16/24	1964	1924	844	627	3474	
FT OD (µm)		22/24	231	264	115	115	526	R^2^ = 0.603
FT OS (µm)		21/24	225	233	109	36	567	

Abbreviations: CME = cystoid macular edema, IHRM = intraretinal hyperreflective material, ERM = epiretinal membrane, EZ = ellipsoid zone, FT = foveal thickness, OD = right eye, and OS = left eye.

## Data Availability

The data presented in this study are available on request from the corresponding author.

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
