# Peer review of "Clinical Phenotype of PDE6B-Associated Retinitis Pigmentosa"

_ijms, 2021, doi:10.3390/ijms22052374_

Round 1

Reviewer 1 Report

Kuehlewein et al. report PDE6B variants in 24 patients from 21 families as well as accompanying clinical manifestations. It's one of the largest PDE6B cohorts described so far in the literature. Thirteen of the sequence variants had been published previously in different studies and 10 variants seem to be novel. In 15 of the 24 patients, the variant appeared homozygous while nine patients revealed two heterozygous variants. Segregation analysis was performed in 6 families out of the 21. In 19 out of the 21 index patients, IRD gene panel sequencing was performed. The remaining two index cases were analyzed by WGS and candidate gene sequencing.

The numbers are inconsistent throughout the manuscript. This is puzzling and confusing while reading. 

Line 35: 50% of variants are novel; lines 86-88: 10 were novel and 13 reported previously (which is 43% novel and not 50%).

Lines 90-94: It is not clear how many families or couples were available for segregation analysis. According to line 375, segregation analysis was carried out only in 6 out of the 21 families. Thus, biallelic inheritance of the variants, either homozygous or compound heterozygous, cannot be claimed in cases where segregation analysis could not be performed. Tables 1 and 2 should contain this information. Columns in Table 1 should not be labelled 'allele 1' and 'allele 2' in case of lacking segregation data but rather 'variant 1' and 'variant 2' when it is not clear if the variants occur in cis or in trans, which is the case for the majority of families (15 out of 21).

Line 200: ‘… 30 individuals carried likely biallelic PDE6B variants …’ contradicts the patient number 24 (from 21 families) as referred to on line 29 and 20 on line 31.

In 15 out of the 24 patients only one (homozygous?) variant has been detected, which raises the question if both alleles are carrying the same variant or if the variant is hemizygous due to a deletion on the second allele. Again, this can only be determined in cases with segregation analysis performed with samples from the parents.

CNV analysis is not mentioned in the manuscript but it would be interesting to learn if this has also been looked at by the authors. IRD gene panel sequencing (19 cases) as well as WGS (1 case) should deliver the relevant data for CNV analysis.

With regard to the clinical manifestations it was surprising to read that none of the patients showed clinical features of CSNB (lines 302-303). Usually, night blindness is an early onset disease manifestation also in RP and I was wondering if this is different in RP patients with PDE6B variants.

Minor:

Not all novel variants from this study have been submitted to ClinVar. This should be done and may be mentioned in the manuscript.

Table 2, column 'Location of missense variant in polypeptide': polypeptide might better be replaced by 'protein domain'

Reviewer 2 Report

It is a  retrospective, longitudinal, observational cohort study, with phenotype genotype correlation of retinitis pigmentosa associated with variants in the PDE6B gene. 

the paper is well illustrated. 

- In terms of FAF results is there any correlation between AF ring dimensions, EZ width and VF, please add it to the results. 

- Could you add a figure illustrating different Full-field electroretinography (ff-ERG) findings in your cohort. 

- Methods:  please provide more details about OCT parameters  measurements 

- Which test was used for studing correlation between EZ width and VF. PLease add details about statistic study in methods section. 

- In table 1 please respect the same type of righting in the nomenclature: 

c.[299G>A;1401+4_1401+48del]; p.[(R100H);?] c.2326G>A;p.(D776N) - p.? not correct --->  right p.(?) - In the discussion also:  (p.[(R100H);(?)]  (p.[(G137R);(?)]) - If you right like this the variant  c.[409G>A;928-9_940dup] please keep that in all the text and table - in table 2 please add also refrence for ACMG prediction and ACMG classification  

Round 2

Reviewer 1 Report

no further comments